# Containment-Formation Control for Second-Order Nonlinear Multi-Agent Systems via Event-Triggering

1st Xinyang Lan
*School of Electrical Engineering*
*Liaoning University of Technology*
Jinzhou, China
xinyang_lan@163.com

2nd Yang Yu
*School of Electrical Engineering*
*Liaoning University of Technology*
Jinzhou, China
am_yuyang@163.com

3rd Wei Wang
*School of Electrical Engineering*
*Liaoning University of Technology*
Jinzhou, China
lgwangw@gmail.com

*Abstract*—This paper studies the event-triggered containment-formation control problem for nonlinear multi-agent systems. Neural networks are utilized to approximate uncertain dynamics. An adaptive neural network controller is designed, and an event-triggering mechanism based on a relative threshold is constructed, which is set on the control channel. By using Lyapunov stability theory, the bounded stability of the closed-loop systems is proven, and the formation tracking error converges to a small neighborhood around zero. It is also shown that the event-triggered control method used does not cause Zeno behavior. Finally, simulation results are provided to verify the effectiveness of the proposed formation control method.

*Index Terms*—Containment-formation, nonlinear multi-agent systems, event-trigger control, adaptive neural network control

## I. INTRODUCTION

In recent years, formation control has been widely used in various fields, such as mobile robots [1], spacecrafts [2] and unmanned aerial vehicles [3]. And the application of formation control allows agents to share information and tasks to achieve collaborative work. The principle of formation control is to design a control strategy to make the intelligences work according to the envisaged formation [4], and this aspect is a hot topic in control research.

Nowadays, there are many formation control methods, including consensus-based formation control, rule-based formation control, behavior-based formation control, and so on. In [5], a consensus algorithm was designed to address formation control problems by appropriately selecting information states upon which consensus was achieved. The auther in [6] considered employing a rule-based strategy to address the unique estimation issues associated with spacecraft formation flying. The aricle in [7] discussed a distributed formation control for networked multi-agent systems. The paper [8] adopted a simple leader-follower formation tracking control strategy to address the formation control problem of non-holonomic mobile robots under communication constraints.

In [9], the authers proposed the application of behavior-based formation control, which achieves the formation control solely by utilizing the relative position information between robots and their neighbors as well as obstacles. And in [10] proposed a communication structure design algorithm that enables followers to form a predetermined formation based on containment control. The formation control mentioned in the above article belongs to the category of formation control with no leader and with one leader. When multi leaders exist in the multi-agent systems, the formation problem is called as containment-formation control. For example, the authers in [11] studied the time-varying output formation for containment control of linear homogeneous and heterogeneous systems. In the [12], the authers proposed a communication structure design algorithm that enables followers to form a predetermined formation based on containment control. Therefore, studying containment-formation control is meaningful.

Due to the limited communication network resources in multi-agent systems, designing event-triggered mechanisms to reduce communication overhead and enhance system efficiency is crucial for ensuring optimal system performance [13]. Consequently, investigating the application of event-triggered control in multi-agent systems is highly significant, as it offers a promising approach to optimizing resource utilization and achieving better overall system performance. Neural networks are used to approximate uncertain dynamics. An adaptive neural network controller is designed, incorporating an event-triggering mechanism based on a relative threshold set on the control channel. Using Lyapunov stability theory, the bounded stability of the closed-loop systems is proven, ensuring the formation tracking error converges to a small neighborhood around zero. It is also demonstrated that the event-triggered control method does not induce Zeno behavior. Simulation results confirm the effectiveness of the proposed formation control method.

This work was supported in part by the National Natural Science Foundation of China under Grant 62273170, and in part by the National Natural Science Foundation of Liaoning Province under Grant 2023JH2/101300187, Grant JYTZD2023082, and Grant 2023-MS-300.

## II. PRELIMINARIES AND PROBLEM FORMULATION

### A. Preliminaries

$\mathcal{F} = \{\mathcal{V}, \mathcal{W}, \mathcal{A}\}$ represents a weighted undirected graph of A orders, where $\mathcal{V} = \{v_1, v_2, \ldots, v_A\}$ is the note set, $\mathcal{W} \subseteq \{(v_i, v_j) : v_i, v_j \in \mathcal{V}\}$ is the edge set, and $\mathcal{A} = [s_{ij}] \subseteq \mathbb{R}^{A \times A}$ is the weighted adjacency matrix. Suppose that there are A agents, where B of them are followers and $A - B > 0$ are leaders. Let $F = \{1, 2, \ldots, B\}$ be the follower subscript set, and $E = \{B+1, B+2, \ldots, A\}$ be the leader subscript set. For any $i, j \in E \cup F$, we can denote $s_{ij}$

$$s_{ij} = \left\{ \begin{array}{ll} 0, & i = j \text{ or } v_{ij} \notin \mathcal{W} \\ b_j > 0, & j \in E \text{ and } v_{ij} \in \mathcal{W} \\ c_{ij} > 0, & j \notin E \text{ and } v_{ij} \in \mathcal{W} \end{array} \right\} \quad (1)$$

*Lemma 1* [14]: The following inequality holds for any $\varepsilon > 0$ and $x \in \mathbb{R}^m$

$$0 \leq x^T \operatorname{sign}(x) - x^T \operatorname{ta}_\varepsilon(x) \leq \varepsilon_1 \quad (2)$$

where $\operatorname{ta}_\varepsilon(x_i) = x_i / \sqrt{x_i^2 + \varepsilon}$, $\operatorname{ta}_\varepsilon(x) = [\operatorname{ta}_\varepsilon(x_1), \operatorname{ta}_\varepsilon(x_2), \ldots, \operatorname{ta}_\varepsilon(x_B)]^T$, $\varepsilon_1 = m\varepsilon$, and sign is the meaning of the sign function.

*Lemma 2* [15]: For any vector $\varepsilon_3 > 0$ and $x \in \mathbb{R}$, the following inequality satisfies

$$0 \leq |x| - x \tanh\left(\frac{x}{\varepsilon_3}\right) \leq 0.2785\varepsilon_3. \quad (3)$$

### B. Problem Description

Consider multi-agent systems consisting of $A$ agents, where $B$ of them are followers and $A - B > 0$ of them are leaders. Let $F = \{1, 2, \ldots, B\}$ be the follower subscript set, and $E = \{B+1, B+2, \ldots, A\}$ be the leader subscript set.

The dynamics of follower $i$ and the leader $k (k \in E)$ can be modeled as follows

$$\left\{ \begin{array}{l} \dot{x}_i(t) = v_i(t) + q_i(x_i(t), t) + \Psi_{1i}(t) \\ \dot{v}_i(t) = s_i(x_i(t), v_i(t), t) + u_i(t) + \Psi_{2i}(t) \end{array} \right. \quad (4)$$

$$\left\{ \begin{array}{l} \dot{x}_k(t) = v_k(t) \\ \dot{v}_k(t) = u_k(t) \end{array} \right. \quad (5)$$

where $i = 1, 2, \ldots, B$, $k = B+1, B+2, \ldots, A$, $x_i(t)$, $v_i(t)$ are the states input vectors of followers, while $v_k(t)$ are the states input vectors of leaders. $u_i(t)$, $u_k(t)$ represent the control vectors of system. $q_i(x_i(t), t)$ and $s_i(x_1(t), v_i(t), t)$ are the uncertain dynamics. $\Psi_{1i}(t)$ and $\Psi_{2i}(t)$ are the unknown disturbances.

*Defmition* 1 : A time-varying formation is specified by a vector $c_F(t) = [c_1^T(t), c_2^T(t), \ldots, c_B^T(t)]^T \in \mathbb{R}^{2B}$ with $c_i(t) = [c_{ix}(t), c_{iv}(t)]^T (i \in F)$ piecewise continuously differentiable. For the multi-agent system (3) and (4), the containment formation is achieved, if for any given bounded initial states $x_i(0)$ and $v_i(0)$ the following conditions are satisfied

1) All the states of the $i$th closed-loop system of the $i$ th follower are bounded.

2) There is positive scalar $\varepsilon_1$ and $\varepsilon_2$, a finite time $T_\varepsilon(\forall t \geq T_\varepsilon)$ such that

$$\begin{array}{l} \|x_i(t) - c_{ix}(t) - x_k(t)\| \leq \varepsilon_1 \\ \|v_i(t) - c_{iv}(t) - v_k(t)\| \leq \varepsilon_2 \end{array} \quad (6)$$

where $\varepsilon_1$ is called the tracking position error, and $\varepsilon_2$ is called the tracking velocity error.

*Assumption* 1: All the following containment formation in this paper must satisfy the time-varying formation condition

$$\lim_{t \to \infty} (c_v(t) - \dot{c}_x(t)) = 0 \quad (7)$$

From the models of system, we get that the Laplacian matrix $L$ can be described as follows

$$L = \begin{bmatrix} L_1 & L_2 \\ \mathbf{0} & 0 \end{bmatrix} \quad (8)$$

where $L_1 = [s_{ij}] \in \mathbb{R}^{B \times B}$ and $L_2 = [s_{i,k}] \in \mathbb{R}^{B \times (A-B)}$. Let $\partial_{xi}(t) = \sum_{j=1}^B s_{ij}((x_i(t) - c_{ix}(t)) - (x_j(t) - c_{jx}(t))) + \sum_{k=B+1}^A s_{ik}(x_i(t) - c_{ix}(t) - x_k(t))$, $\partial_{vi}(t) = \sum_{j=1}^B s_{ij}((v_i(t) - c_{iv}(t)) - (v_j(t) - c_{jv}(t))) + \sum_{k=B+1}^A s_{ik}(v_i(t) - c_{iv}(t) - v_k(t))$ be the two auxiliary variables of the multi-agent system.

Let $\partial_x(t) = [\partial_{x1}^T(t), \partial_{x2}^T(t), \ldots, \partial_{xB}^T(t)]^T$, $\partial_v(t) = [\partial_{v1}^T(t), \partial_{v2}^T(t), \ldots, \partial_{vB}^T(t)^T]$, $u(t) = [u_1^T(t), u_2^T(t), \ldots, u_B^T(t)]^T$, $c_v(t) = [c_{1v}^T(t), c_{2v}^T(t), \ldots, c_{Bv}^T(t)]^T$, $c_x(t) = [c_{1x}^T(t), c_{2x}^T(t), \ldots, c_{Bx}^T(t)]^T$, $\dot{c}_x(t) = [\dot{c}_{1x}^T(t), \dot{c}_{2x}^T(t), \ldots \dot{c}_{Bx}^T(t)]^T$, and $u_E(t) = [u_1(t), u_2(t), \ldots, u_B(t)]^T$.

Then, denoting $\zeta(t) = [x(t), v(t)]^T$, multi-agent systems can be written in the following compact form

$$\left\{ \begin{array}{l} \dot{\partial}_x(t) = \partial_v(t) + L_1 q(x(t), t) \\ \qquad + L_1(c_v(t) - \dot{c}_x(t)) + L_1 \Psi_1(t), \\ \dot{\partial}_v(t) = L_1(s(\zeta(t), t) + u(t) + \Psi_2(t)) + L_2 u_E(t). \end{array} \right. \quad (9)$$

where $q(x(t), t) = [q_1(x_1(t), t), q_2(x_2(t), t) \ldots, q_B(x_B(t), t)]^T$, $f(\zeta(t), t) = [f_1(\zeta_1(t), t), f_2(\zeta_2(t), t), \ldots, f_B(\zeta_B(t), t)]^T - \dot{c}_v(t)$, and $\Psi_z(t) = [\Psi_{z1}(t), \Psi_{z2}(t), \ldots, \Psi_{zB}(t)]^T (z = 1, 2)$.

## III. CONTROLLER AND EVENT TRIGGERING MECHANISM DESIGN

In this section, the containment-formation controller will be carried out based on the dynamic surface control design. And the event-triggering mechanism based on a relative threshold will be constructed on the control channel.

Due to $q(x(t), t)$ and $s(\zeta(t), t)$ are uncertain functions, they can be approximated by using the neural networks. Denote $q(x(t), t) = W_1^{*T}(t) S_1(t) + \varepsilon_4$ and $s(\zeta(t), t) = W_s^{*T}(t) S_2(t) + \varepsilon_5$, we can use $\hat{W}_i^T$ to approxiamte $W_i^{*T}$.

Then, multi-agent systems (9) can be written

$$\left\{ \begin{array}{l} \dot{\partial}_x(t) = \partial_v(t) + L_1(W_1^{*T}(t) S_1(t) + \varepsilon_4) \\ \qquad + L_1(c_v(t) - \dot{c}_x(t)) + L_1 \Psi_1(t), \\ \dot{\partial}_v(t) = L_1(W_2^{*T}(t) S_2(t) + u(t)) + L_1(\varepsilon_5 + \Psi_2(t)) \\ \qquad + L_2 u_E(t). \end{array} \right. \quad (10)$$

To design the controller, the error surface can be designed

$$Z_1(t) = \partial_x(t) \tag{11}$$
$$Z_2(t) = \partial_v(t) - \bar{\alpha} \tag{12}$$

where $Z_1(t) = \left[ Z_{11}^T, Z_{12}^T, \ldots, Z_{1B}^T \right]^T$ is tracking error. $\bar{\alpha} = \left[ \bar{\alpha}_1^T, \bar{\alpha}_2^T, \ldots, \bar{\alpha}_B^T \right]^T$ is the virtual controller going to be designed in the following contents.

Firstly, we can take the derivative of the $Z_1(t)$, and combine with $\dot{\partial}(t)$ in (10). And we can denote $L_1 = D + E$, where $D = \text{diag}[L_{1,11}, L_{1,22}, \ldots, L_{1,BB}]$ is the positive definite diagonal matrix of $L_1$.

Thus, we can obtain the following expressions

$$\dot{Z}_1(t) = \partial_v(t) + D(W_1^{*T}(t)S_1(t)) + E(W_1^{*T}(t)S_1(t)) + L_1(\varepsilon_4 + \Psi_1(t)) + L_1(c_v(t) - \dot{c}_x(t)). \tag{13}$$

Since $E(W_1^{*T}(t)S_1(t))$ and $L_1\Psi_1(t)$ are bounded, we can found a positive constant $\gamma_1$ which satisfies the following inequality

$$\left\| E(W_1^{*T}(t)S_1(t)) + L_1\Psi_1(t) \right\| \leq \gamma_1 \tag{14}$$

Building the first Lyapunov function as follows

$$V_1(t) = \frac{1}{2} Z_1^T(t) Z_1(t) \tag{15}$$

Then, the derivative of $V_1(t)$ can be obtained

$$\dot{V}_1(t) = Z_1^T(t)(\dot{Z}_2(t)) = Z_1^T(t)(Z_2(t) + \bar{\alpha} + D(W_1^{*T}(t)S_1(t)) + E(W_1^{*T}(t)S_1(t)) + L_1(\varepsilon_4 + \Psi_1(t)) + L_1(c_v(t) - \dot{c}_x(t))) \tag{16}$$

Considering the Young's inequality, we can get

$$Z_1^T(t)L_1(c_v(t) - \dot{c}_x(t)) \leq \delta_0 Z_1^T(t)Z_1(t) + \frac{1}{4\delta_0} \|L_1\|^2 \|c_v(t) - \dot{c}_x(t)\|^2 \tag{17}$$
$$Z_1^T(t)L_1\varepsilon_4 \leq \delta_1 Z_1^T(t)Z_1(t) + \frac{\varepsilon_{4,A}^2}{4\delta_1}$$

where $\delta_0$ and $\delta_1$ are all positive scalars, and $\varepsilon_4 \leq \varepsilon_{4,A}$.

Taking (14) and (17) into (16), we can get an inequality

$$\dot{V}_1(t) \leq Z_1^T(t)(Z_2(t) + D(W_1^{*T}(t)S_1(T) + \gamma_1 + \bar{\alpha}_1) + \delta_0 Z_1^T(t)Z_1(t) + \frac{1}{4\delta_0} \|L_1\|^2 \|c_v(t) - \dot{c}_x(t)\|^2 + \delta_1 Z_1^T(t)Z_1(t) + \frac{\varepsilon_{4,A}^2}{4\delta_1} \tag{18}$$

For the above inequality, the virtual controller $\alpha_1$ can be cosntructed

$$\alpha_1(t) = -C_1 Z_1(t) - D\hat{W}_1^T(t)S_1(t) - \gamma_1 \text{ta}_\varepsilon(Z_1(t)) \tag{19}$$

To avoid taking the derivative of the above virtual controller, the following low-pass filter is introduced as

$$T\dot{\bar{\alpha}}_1(t) + \bar{\alpha}_1(t) = \alpha_1(t) \tag{20}$$

where $T$ is a positive time contant.

Denote $\sigma(t) = \bar{\alpha}_1(t) - \alpha_1(t)$, the above equation can be rewritten

$$\dot{\sigma}(t) = -T^{-1}\sigma(t) - \dot{\alpha}_1(t). \tag{21}$$

Taking (19) and (20) into (18), we can get

$$\dot{V}_1(t) \leq Z_1^T(t)(Z_2(t) - C_1 Z_1(t) + D(\tilde{W}_1^T(t)S_1(T) + \gamma_1 - \gamma_1 \text{ta}_\varepsilon(Z_1(t) + \sigma(t)) + \delta_0 Z_1^T(t)Z_1(t) + \frac{1}{4\delta_0} \|L_1\|^2 \|c_v(t) - \dot{c}_x(t)\|^2 + \delta_1 Z_1^T(t)Z_1(t) + \frac{\varepsilon_{4,A}^2}{4\delta_1} \tag{22}$$

where $\tilde{W}_1^T(t) = W_1^{*T} - \hat{W}_1^T$.

By the $Lemma$ 1, we can get

$$-\gamma_1 Z_1^T(t)\text{ta}_\varepsilon(Z_1(t)) + Z_1^T \left( E\left( W_1^T(t)S_1(t) \right) + L_1\Psi_1(t) \right) \leq \gamma_2 \tag{23}$$

According to the Young's inequality, we can obtain that

$$Z_1^T(t)\sigma(t) \leq \frac{\delta_2}{2} Z_1^T(t)Z_1(t) + \frac{1}{2\delta_2} \sigma^T(t)\sigma(t) \tag{24}$$

where $\delta_2$ is a positive scalar.

Define $\gamma_2 = \gamma_1 + \frac{\varepsilon_{4,A}}{4\delta_1}$.

Then, taking (23) and (24) into (22), we can get

$$\dot{V}_1(t) \leq Z_1^T(t)(Z_2(t) - C_1 Z_1(t) + D(\tilde{W}_1^T(t)S_1(T)) + \gamma_2 + \delta_0 Z_1^T(t)Z_1(t) + \frac{1}{4\delta_0} \|L_1\|^2 \|c_v(t) - \dot{c}_x(t)\|^2 + \frac{\delta_2}{2} Z_1^T(t)Z_1(t) + \frac{1}{2\delta_2} \sigma^T(t)\sigma(t) + \delta_1 Z_1^T(t)Z_1(t) + \frac{\varepsilon_{4,A}^2}{4\delta_1} \tag{25}$$

Secondly, the derivative of $Z_2(t)$ can be obtained

$$\dot{Z}_2(t) = \dot{\xi}_v(t) - \dot{\bar{\alpha}}_1(t) = L_1 \left( W_2^{*T}(t)S_2(t) + \varepsilon_5 + u(t) + \Psi_2 \right) + L_2 u_E(t) - \dot{\bar{\alpha}}_1(t). \tag{26}$$

Since $\varepsilon_5$, $L_1^{-1}L_2 u_E(t)$, $\Psi_2$ and $L_1^{-1}\dot{\bar{\alpha}}_1(t)$ are bounded, we can obtain that

$$\left\| (\Psi_2(t) + \varepsilon_5 + L_1^{-1}L_2 u_E(t) - L_1^{-1}\dot{\bar{\alpha}}_1(t))_i \right\|_\infty \leq \gamma_{3i} \tag{27}$$

We can construct $u(t)$

$$u(t) = -C_2 Z_2(t) - Z_1(t) - \hat{W}_2^T(t)S_2(t) - \hat{\gamma}_3 \tag{28}$$

Due to the event-triggering mechanism shuld be set on the control channel, the event-triggering can be built

$$o(t) = u(t_k), \quad \forall t \in [t_k, t_{k+1}) \tag{29}$$
$$t_{k+1} = \inf\{t > t_k | |e(t)| \geq \vartheta |o(t)| + m\}$$

where $e(t) = u(t) - o(t)$ represents the measurement error casued by event trigger, $0 < \vartheta < 1$, $\varepsilon, \sigma, m$ are all positive constants, and $\tilde{m} > \frac{m}{1-\vartheta}.t_k, k \in z^+$, represents the moment when the event is triggered.

That is when (22) be sparked while this moment marks as $t_k$, and the control signal $u(t_{k+1})$ of this moment can apply to the system. In the gap between two event-triggered moments $t \in [t_k, t_{k+1})$, the control signal $u(t_k)$ remains unchanged.

Denote $l(t) = -\hat{W}_2^T(t)S_2(t) - C_2 Z_2(t) - Z_1(t) - \hat{\gamma}_3(t)$.

Thus, $u(t)$ can be rewritten

$$u(t) = -(1+\vartheta)[l(t)]tanh\left(\frac{Z_2(t)(l(t))}{\varepsilon_3}\right)$$
$$+ \tilde{m}\tanh(\frac{Z_2(t)\tilde{m}}{\varepsilon_3}) \tag{30}$$

Triggering inerval between two events $[t_k, t_{k+1})$, (23) can be obtained

$$|o(t) - u(t)| \leq \vartheta |o(t)| + m \tag{31}$$

$$u(t) = (1 + \rho_1(t)\vartheta)o(t) + \rho_2(t)m \tag{32}$$

where $|\rho_1(t)| \leq 1$ and $|\rho_2(t)| \leq 1$.

Then, we can obtain that

$$o(t) = \frac{u(t)}{1+\rho_1(t)\vartheta} - \frac{\rho_2(t)m}{1+\rho_1(t)\vartheta} \tag{33}$$

Consider the second Lyapunov function

$$V_2(t) = \frac{1}{2}Z_2^T(t)Z_2(t) + \frac{1}{2}\sum_{i=1}^{B}(\hat{\gamma}_{3i} - \gamma_{3i})^2/\varsigma_0 \tag{34}$$

Then, substituting $u(t)$ into (26) yields

$$\dot{Z}_2(t) = L_1(W_2^{*T}(t)S_2(t) + \varepsilon_5$$
$$+ \frac{u(t)-\rho_2(t)m}{1+\rho_1(t)\vartheta} + \Psi_2) + L_2 u_E - \dot{\alpha}_1(t) \tag{35}$$

Taking the derivative of $V_2(t)$ yields

$$\dot{V}_2(t) = Z_2^T(t)(W_2^{*T}(t)S_2(t) + \varepsilon_5 + \frac{u(t)-\rho_2(t)m_1}{1+\rho_1(t)\vartheta} + \Psi_2)$$
$$+ Z_2^T(t)(L_1^{-1}L_2 u_E - L_1^{-1}\dot{\alpha}_1(t)) + \sum_{i=1}^{B}(\hat{\gamma}_{3i} - \gamma_{3i})\dot{\hat{\gamma}}_{3i}/\varsigma_0$$
$$+ \sum_{i=1}^{B}(\hat{\gamma}_{3i} - \gamma_{3i})|Z_2^T(t)| \qquad . \tag{36}$$

Due to $\lambda_1(t) \in [-1, 1], \lambda_2(t) \in [-1, 1]$, we can obtain

$$\frac{z_2 u}{1+\rho_1(t)\vartheta} \leq \frac{z_2 u}{1+\vartheta}$$
$$\left|\frac{\rho_2(t)m}{1+\rho_1(t)\vartheta}\right| \leq \frac{m}{1-\vartheta} \tag{37}$$

According to $Lemma\ \ 2$, we can get

$$Z_2^T(t)(\frac{u(t)-\rho_2(t)m}{1+\rho_1\vartheta})$$
$$\leq Z_2^T(t)(-l(t)\tanh(\frac{Z_2(t)l(t)}{\varepsilon_3}) + \tilde{m}\tanh(\frac{Z_2\tilde{m}}{\varepsilon_3})) \tag{38}$$
$$\leq 0.557\varepsilon_3 + Z_2^T(t)l(t)$$

Then, we can obtain

$$-\sum_{i=1}^{M}\gamma_{3i}|Z_{2i}^T(t)|$$
$$+ Z_2^T(t)(\Psi_2(t) + \varepsilon_5 + L_1^{-1}L_2 u_E(t) - L_1^{-1}\dot{\alpha}(t)) \tag{39}$$
$$\leq -\sum_{i=1}^{B}\gamma_{2i}|Z_{2i}^T(t)| + \sum_{i=1}^{B}\gamma_{3i}|Z_{2i}^T(t)| = 0.$$

Noting that $\dot{\hat{\gamma}}_{2i}/\varsigma_0 + |Z_2^T(t)| = 0$, and substituting $l(t)$ into (36), we can obtain

$$\dot{V}_2(t) \leq -Z_2^T(t)C_2 Z_2(t) - Z_2^T(t)Z_1(t)$$
$$+ Z_2^T(t)\tilde{W}_2^T(t)S_2(t) + 0.557\varepsilon_3. \tag{40}$$

## IV. STABILITY ANALYSIS

*Theorem 1*: For nonlinear second-order systems (4) and (5), if the controller (28), virtual controller (19), and the adaptive laws (28), (21), (46) and (47) are applied. Then, all signals in the closed-loop system are bounded and the formation tracking error will converge to a small range.

To prove the Theorem 1, we can consider $V(t)$

$$V(t) = V_1(t) + V_2(t) + \frac{1}{2}\sigma^T(t)\sigma(t)$$
$$+ \frac{1}{2}\text{tr}\left[\tilde{W}_1^T(t)\varepsilon_6^{-1}\tilde{W}_1(t)\right] \tag{41}$$
$$+ \frac{1}{2}\text{tr}\left[\tilde{W}_2^T(t)\varepsilon_7^{-1}\tilde{W}_2(t)\right].$$

By(22)-(40), taking the derivative of $V(t)$, we can get

$$\dot{V}(t) = \dot{V}_1(t) + \dot{V}_2(t) + \sigma^T(t)\dot{\sigma}(t)$$
$$+ tr[\tilde{W}_1^T(t)\varepsilon_6^{-1}\dot{W}_1(t)] + tr[\tilde{W}_2^T(t)\varepsilon_7^{-1}\dot{W}_2(t)]$$
$$\leq -Z_1^T(t)\left(C_1 - \left(\frac{\delta_2}{2} + \delta_0 + \delta_1\right)I_2\right)Z_1(t)$$
$$- Z_2^T(t)C_2 Z_2(t) + 0.557\varepsilon_3 + \frac{1}{2\delta_2}\sigma^T(t)\sigma(t)$$
$$+ \sigma^T(t)\dot{\sigma}(t) + Z_1^T(t)D\tilde{W}_1^T(t)S_1(t) + Z_2^T(t)\tilde{W}_2^T(t)S_2(t)$$
$$- \text{tr}\left[\tilde{W}_1^T(t)\varepsilon_6^{-1}\dot{\hat{W}}_1(t)\right] - \text{tr}\left[\tilde{W}_2^T(t)\varepsilon_7^{-1}\dot{\hat{W}}_2(t)\right]$$
$$+ \gamma_2 + \frac{1}{4\delta_0}\|L_1\|^2\|c_v(t) - \dot{c}_x(t)\|^2. \tag{42}$$

From (21), it follows that

$$\sigma^T(t)\dot{\sigma}(t) = -T^{-1}\sigma(t) - \sigma^T(t)\dot{\alpha}_1(t)$$
$$\leq -\sigma^T(t)\left(T^{-1} - \frac{\delta_4}{2}I\right)\sigma(t) + \frac{1}{2\delta_4}\|\dot{\alpha}_1(t)\|^2 \tag{43}$$

where $\delta_4$ is a positive scalar.By the property of the trace of a matrix, which has $Z_1^T(t)D\tilde{W}_1^T(t)S_1(t) = tr[\tilde{W}_1^T(t)S_1(t)Z_1^T(t)D]$ and $Z_2^T(t)\tilde{W}_2^T(t)S_2(t) = tr[\tilde{W}_1^T(t)S_2(t)Z_2^T(t)]$.

Substituting (38) into (37), we can obtain

$$\dot{V}(t) \leq -Z_1^T(t)\left(C_1 - \left(\frac{\delta_2}{2} + \delta_0 + \delta_1\right)I_2\right)Z_1(t)$$
$$- Z_2^T(t)C_2 Z_2(t) + 0.557\varepsilon_3$$
$$- \sigma^T(t)\left(T^{-1} - \frac{1}{2\delta_2} - \frac{\delta_4}{2}I\right)\sigma(t)$$
$$+ \text{tr}\left[\tilde{W}_1^T(t)\left(S_1(t)Z_1^T(t)D - \varepsilon_6^{-1}\dot{\hat{W}}_1(t)\right)\right]$$
$$+ \text{tr}\left[\tilde{W}_2^T(t)\left(S_2(t)Z_2^T(t) - \varepsilon_7^{-1}\dot{\hat{W}}_2(t)\right)\right]$$
$$+ \frac{1}{2\delta_4}\|\dot{\alpha}_1(t)\|^2 + \gamma_2 + \frac{1}{4\delta_0}\|L_1\|^2\|c_v(t) - \dot{c}_x(t)\|^2. \tag{44}$$

We can get

$$\dot{\hat{W}}_1(t) = \varepsilon_6 S_1(t) Z_1^T(t) D - \varepsilon_6 \Xi_1(t) \hat{W}_1(t) \quad (45)$$

$$\dot{\hat{W}}_2(t) = \varepsilon_7 S_2(t) Z_2^T(t) - \varepsilon_7 \Xi_2(t) \hat{W}_2(t). \quad (46)$$

So taking (45) and (46) into (44), the equation about $\tilde{W}_1$ changes

$$\text{tr}\left[\tilde{W}_1^T(t)\left(S_1(t) Z_1^T(t) D - \varepsilon_6^{-1}\dot{\hat{W}}_1(t)\right)\right]$$
$$= \text{tr}\left[\tilde{W}_1^T(t)\Xi_1(t)\hat{W}_1(t)\right] \quad (47)$$
$$\leq -\varphi_1 \text{tr}\left[\tilde{W}_1^T(t)\tilde{W}_1(t)\right] + \frac{\lambda_{\max}(\Xi_1)}{2\mu_1}\|W_1^*(t)\|_F^2$$

where $0 \leq \mu_1 \leq \frac{2\lambda_{\min}(\Xi_1)}{\lambda_{\max}(\Xi_1)}$, $\varphi_1 = \left(\lambda_{\min}(\Xi_1) - \frac{1}{2}\mu_1\lambda_{\max}(\Xi_1)\right)$.

By the same reasoning, we can build $\tilde{W}_2$.

From (44), we can get

$$\dot{V}(t) \leq -Z_1^T(t)\left(C_1 - (\frac{\delta_2}{2} + \delta_0 + \delta_1)I_2\right) Z_1(t)$$
$$- Z_2^T(t) C_2 Z_2(t) + 0.557\varepsilon_3$$
$$- \sigma^T(t)(T^{-1} - \frac{1}{2\delta_2} - \frac{\delta_4}{2})\sigma(t)$$
$$- \varphi_1 tr[\tilde{W}_1^T(t)\tilde{W}_1(t)] - \varphi_2 tr\left[\tilde{W}_2^T(t)\tilde{W}_2(t)\right]$$
$$+ \frac{\lambda_{\max}(\Xi_1)}{2\mu_1}\|W_1^*(t)\|_F^2 + \frac{\lambda_{\max}(\Xi_2)}{2\mu_2}\|W_2^*(t)\|_F^2$$
$$+ \frac{1}{2\delta_4}\|\dot{\alpha}_1(t)\|^2 + \gamma_2 + \frac{1}{4\delta_0}\|L_1\|^2\|c_v(t) - \dot{c}_x(t)\|^2$$
$$\leq -\alpha_0 V + \beta_1 + \frac{1}{4\delta_0}\|L_1\|^2\|c_v(t) - \dot{c}_x(t)\|^2$$
$$(48)$$

where $\beta_1 = \|\dot{\alpha}_1(t)\|^2/2\delta_4 + \gamma_2 + 0.557\varepsilon_3 + \lambda_{\max}(\Xi_1)\|W_1^*(t)\|_F^2/(2\mu_1) + \lambda_{\max}(\Xi_2)\|W_2^*(t)\|_F^2/(2\mu_2)$, $\alpha_0 = \min\{\lambda_1, \lambda_2, \lambda_3, \lambda_4, \lambda_5\}$, $\lambda_1 = -\lambda_{\min}(C_1 - (\delta_2/2 + \delta_0 + \delta_1)I)$, $\lambda_2 = -\lambda_{\min}(C_2)$, $\lambda_3 = -\lambda_{\min}(\Xi^{-1} - (1/2 + \delta_4/2)I)$, $\lambda_4 = \varphi_1$ and $\lambda_5 = \varphi_2$.

Denote $\Phi_i(t) = (1/(4\delta_0))\|L_1\|^2\|c_{iv}(t) - \dot{c}_{ix}(t)\|^2$ and we can define $\Phi(t) = (1/(4\delta_0))\|L_1\|^2\sum_{i=1}^{A-1}\|c_{iv}(t) - \dot{c}_{ix}(t)\|^2$.

Then, inequality (48) can be rewritten as

$$\dot{V}(t) \leq -\alpha_0 V(t) + \beta_1 + \Phi(t). \quad (49)$$

Then, we should prove the prerequisities before anlysis the system stability.

Firstly, integrating (49) gives

$$V(t) \leq e^{-\alpha_0 t}V(0) + \frac{\beta_1}{\alpha_0}(1 - e^{-\alpha_0 t}) + \int_0^t (e^{-\alpha_0(t-\tau)}\Phi(\tau))d\tau$$
$$= e^{-\alpha_0 t}V(0) + \frac{\beta_1}{\alpha_0}(1 - e^{-\alpha_0 t}) + e^{-\alpha_0 t}\int_0^t (e^{\alpha_0\tau}\Phi(\tau))d\tau.$$
$$(50)$$

One has $\lim_{t\to\infty}\Phi(t) = 0$ and $\Phi(t)$ is uniformly bounded, which means that there is a positive constant $\Phi_0$ such that $\Phi(t) \leq \Phi_0$. Denote a positive constant $\varpi > 0$, there exists a fixed time $T_\varpi > 0$, such that $0 < \Phi(t) < \varpi/4$, $\forall t > T_\varpi$.

Then, we can obtain that

$$V(t) - \frac{\beta_1}{\alpha_0} \leq e^{-\alpha_0 t}V(0) - \frac{\beta_1}{\alpha_0}e^{-\alpha_0 t}$$
$$+ e^{-\alpha_0 t}\Phi_0(e^{\alpha_0 T_\varpi} - 1) \quad (51)$$
$$+ \frac{\varpi}{4}e^{-\alpha_0 t}(e^{\alpha_0 t - e^{\alpha_0 T_\varpi}}).$$

Found a time constant $T_D$ which satisfies $\forall t > T_D > T_\varpi$. One obtains $e^{-\alpha_0 t}V(0) < \varpi/4$, $|-e^{-\alpha_0 t}\beta_0/\alpha_0| < \varpi/4$, $e^{-\alpha_0 t}\Phi_0(e^{\alpha_0 T_\varpi} - 1) < \varpi/4$, and $(1 - e^{-\alpha_0(t-T_\varpi)})\varpi/4 < \varpi/4$.

Thus, (51) can be changed to $\dot{V}(t) \leq \frac{\beta_1}{\alpha_0} + \varpi$, for the reason $\lim_{t\to\infty}\varpi = 0$ and $\lim_{t\to\infty}V(t) \leq \beta_1/\alpha_0$.

We can easily get that from (6) and (8): $\|x_i(t) - c_{ix}(t) - x_k(t)\| = \|L_1^{-1}\partial_{xi}(t)\|$ and $\|v_i(t) - c_{iv}(t) - v_k(t)\| = \|L_1^{-1}\partial_{vi}(t)\|$. It is easy to get that the above equations are bounded.

Due to the introduction of event triggers, Zeno should be prevented. From (23) we can obtain $e(t) = u(t) - o(t)$, $\forall t \in [t_k, t_{k+1})$.

Then, the derivitivate of $e(t)$

$$\frac{d}{dt}|e| = \frac{d}{dt}(e*e)^{\frac{1}{2}} = sign(e)\dot{e} \leq |\dot{u}| \quad (52)$$

Since $\dot{u}$ is a continuous function, and the variables in the function are all globally bounded. So there must be a constant $\chi$ which satisfies $|\dot{u}| \leq \chi$. When $e(t_k) = 0$ and $\lim_{t\to t_{k+1}}e(t) = \vartheta u$, hence a positive constant $t^*$ which satisfies $t^* \geq \frac{\vartheta u}{\chi}$ can be found. Moreover, the Zeno phenomenon does not occur. From all the analysis above, it is concluded that all closed-loop signals of multi-agent systems are bounded. The containment-formation control protocols and event-triggering mechanism are designed.

## V. SIMULATION

Consider multi-agent systems combining six followers and two leaders. The mathematical models of followers can be constructed

$$\begin{aligned}\dot{x}_i(t) &= G_1 * v_i(t) + q_i(x_i(t), t) + \Psi_{1i}(t)\\\dot{v}_i(t) &= G_2 * u_i(t) + s_i(x_i(t), v_i(t), t) + \Psi_{2i}(t)\end{aligned} \quad (53)$$

where $i = 1, 2, \ldots, 6$, $G_1 = 1.5$, $G_2 = 5$, $\Psi_{1i}(t)$ and $\Psi_{2i}(t)$ represent the disturbances. Suppose $\Psi_{1i}(t) = 2 + 0.3\sin(0.2t) + 0.2\cos(2t)$ and $\Psi_{2i}(t) = 2\cos(0.2t)$. And $q_i(x_i(t), t)$ and $s_i(x_i(t), v_i(t), t)$ are the nonlinear parts, we set them as $q_i(x_i(t), t) = d_i cos(x_i(t))^2$ and $s_i(x_i(t), v_i(t), t) = n_i sin(x_i v_i)^2$. The constants $d_i$ and $n_i$ belong to $[-1; 1; 1.5; 2; -1.5; -2]$ and $[2; 0.3; 1.6; ; 0.6; 0.8; 0.4]$, respectively.

The leaders are modeled as $x_k = \begin{bmatrix}\sin(t)^2 + 1\\2\sin(t)^2 + 0.8\end{bmatrix}$, $v_k = \begin{bmatrix}\cos(t)^2 + 0.6\\1.2\cos(t)^2 + 0.5\end{bmatrix}$.

The follwer adjacency matrix is $\begin{bmatrix} 0 & 1 & 0 & 0 & 0 & 1 \\ 1 & 0 & 1 & 0 & 0 & 0 \\ 0 & 1 & 0 & 1 & 0 & 0 \\ 0 & 0 & 1 & 0 & 1 & 0 \\ 0 & 0 & 0 & 1 & 0 & 1 \\ 1 & 0 & 0 & 0 & 1 & 0 \end{bmatrix}$,

and the leader adjacency matrix is $\begin{bmatrix} 1 & 1 \\ 0 & 0 \\ 1 & 1 \\ 1 & 1 \\ 0 & 0 \\ 0 & 0 \end{bmatrix}$.

The desired containment-formation of followers are presented

$$\begin{aligned} c_{xi} &= \begin{bmatrix} 0.8\cos(t + \frac{i-1}{3}\pi) \end{bmatrix} \\ c_{vi} &= \begin{bmatrix} -0.8\sin(t + \frac{i-1}{3}\pi) \end{bmatrix} \end{aligned} \tag{54}$$

The parameters could be chosen $C_1 = 50$, $C_2 = 50$, $T = 0.001$, $\varepsilon_6 = 0.01$, $\varepsilon_7 = 0.01$, $\Xi_1 = 0.005$, $\Xi_2 = 0.005$, $\gamma_1 = 2.5$, $m = 1$, $\tilde{m} = 2$, $\vartheta = 0.5$, $\varepsilon_3 = 0.5$.

The simulation resluts are shown, Figs. 1-4 describe snapshots of the states of six followers and two leaders at t=0s, t=5s, t=15s, and t=20s.

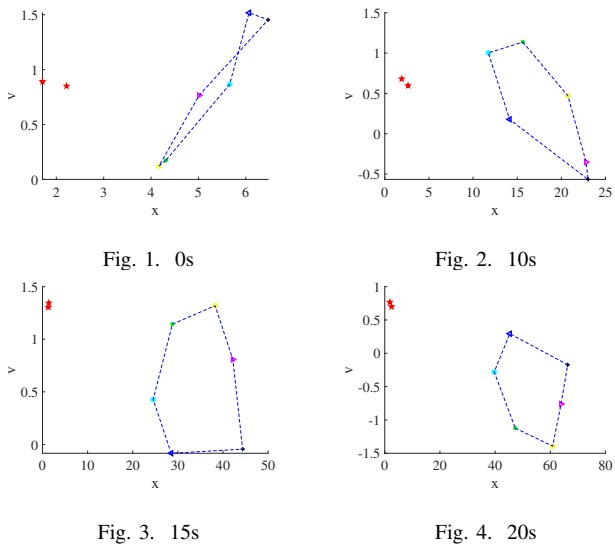

Fig. 1.  0s          Fig. 2.  10s

Fig. 3.  15s          Fig. 4.  20s

Figs. 5-8 are the diagrams of continuous controller, event-triggered controller, and control moments shows that event-triggered control effectively reduces the update frequency of the controller.

## VI. CONCULSIONS

This paper studied the containment-formation control for second-order nonlinear multi-agent systems via event-triggering. An adaptive neural network controller and a relative threshold event-triggered mechanism based on control signals have been designed to control multi-agent systems. In the multi-agent systems, neural networks are used to approximate unknown dynamics. In terms of stability, Lyapunov's method is adopted to prove it. Furthermore, it is demonstrated that there is no Zeno phenomenon in the multi-agent systems.

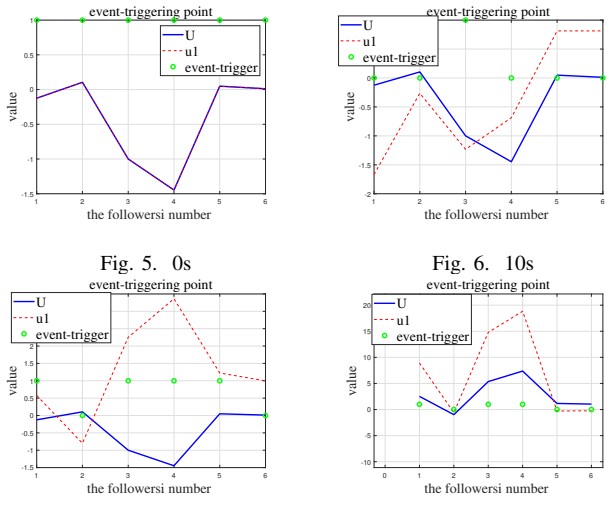

Fig. 5.  0s          Fig. 6.  10s

Fig. 7.  15s          Fig. 8.  20s

Finally, simulation results are presented to validate the effectiveness of the proposed formation control method.

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
