# OpenReview forum: "Containment-Formation Control for Second-Order Nonlinear Multi-Agent Systems via Event-Triggering"
_IEEE.org/ICIST/2024/Conference — IEEE ICIST 2024 Conference Submission_

### Official Review · Reviewer_p5KD · 2024-08-25
**Containment-Formation Control for Second-Order Nonlinear Multi-Agent Systems via Event-Triggering**

**Rating:** 7
**Confidence:** 2

**Review:**

This paper investigates the event-triggered containment control problem in multi-agent systems. The research work is substantial, and although there are some issues, it can be considered for acceptance after revisions.

---

### Official Review · Reviewer_HvNh · 2024-09-01
**Comments to paper 46**

**Rating:** 8
**Confidence:** 5

**Review:**

This paper studies the event-triggered containmentformation control problem for nonlinear multi-agent systems. Neural networks are utilized to approximate uncertain dynamics. Some comments should be considered.
1. The controller has many parameters. Please give a remark to illustrate the effect of the controller
parameters.
2. Which is the training data, which is the validation data?
3. Which is the architecture of the neural network?
4.How does the event triggering strategy proposed in this article differ from previous results?

---

### Official Review · Reviewer_6MWU · 2024-09-01
**Accept**

**Rating:** 8
**Confidence:** 5

**Review:**

1. The authors should explicitly outline the motivation and contributions of this paper in comparison to existing works in the field.
2. Please clarify the reasonability and feasibility of Assumption 1 in this paper.
3. Comparative results with other references should be presented to validate the effectiveness of the proposed method.
4. Typo: Defmition.
5. The presentation quality of this paper requires significant improvement. The authors are advised to carefully review and correct several typographical and grammatical errors within this paper.

---

### Decision · Program_Chairs · 2024-09-06

Accept (Oral)